ecology

rarity, biodiversity change, immigration, invasive species

**Author for correspondence:**
Faith A. M. Jones
e-mail: faith.jones@ubc.ca

# Recent increases in assemblage rarity are linked to increasing local immigration

Faith A. M. Jones[1,2], Maria Dornelas[1,3]
and Anne E. Magurran[1,3]

[1]Centre for Biological Diversity, School of Biology, University of St Andrews, St Andrews, Fife, UK
[2]Faculty of Forestry, The University of British Columbia, Vancouver, British Columbia, Canada
[3]School of Biology, Scottish Oceans Institute, St Andrews, UK

FAMJ, 0000-0001-6571-714X; MD, 0000-0003-2077-7055;
AEM, 0000-0002-0036-2795

As pressures on biodiversity increase, a better understanding of how assemblages are responding is needed. Because rare species, defined here as those that have locally low abundances, make up a high proportion of assemblage species lists, understanding how the number of rare species within assemblages is changing will help elucidate patterns of recent biodiversity change. Here, we show that the number of rare species within assemblages is increasing, on average, across systems. This increase could arise in two ways: species already present in the assemblage decreasing in abundance but with no increase in extinctions, or additional species entering the assemblage in low numbers associated with an increase in immigration. The positive relationship between change in rarity and change in species richness provides evidence for the second explanation, i.e. higher net immigration than extinction among the rare species. These measurable changes in the structure of assemblages in the recent past underline the need to use multiple biodiversity metrics to understand biodiversity change.

## 1. Introduction

One of the few consistent patterns across ecological assemblages is that they contain few common species and many rare species, meaning rare species contribute disproportionately to species richness [1–3]. Because rare species make up a high fraction of assemblage species lists, a better understanding of how the number of rare species within assemblages is changing over time will help elucidate patterns of recent biodiversity change. Rarity is, however, a complex concept. In her seminal work on rarity, Rabinowitz [4] defined seven different types of rarity based around the characteristics of small geographical ranges, specific habitat

preferences and low population numbers. We focus on the local population facet of this definition because species that are locally rare are particularly sensitive to change due to their small population numbers [5]. We ask the question of how rarity is changing in a time when biodiversity is increasingly threatened.

Pressures on biodiversity are increasing at an alarming rate and show no sign of abating [6]. Consequently, the populations of many species are decreasing [7], and low population sizes are a criterion for assessing extinction risk [8]. Local extinction often takes place a considerable time after populations start to decline, leading to an extinction debt where assemblages in the short term retain many doomed species [9]. If local extinctions do not immediately increase despite the populations being on their way to eventual extinction, then these local population declines may be linked to increasing numbers of rare species within assemblages because more populations fall below a threshold of rarity. While rare species may be retained in the short term because of extinction debt, in this scenario population declines could eventually lead to declining local species richness while rarity increases.

Declines in populations are not the only potential influence on numbers of rare species though, because new species can also enter assemblages. Immigration events maintain local species richness when immigration and local extinction events are balanced [10]. However, rates of turnover within assemblages are increasing, which suggests increasing numbers of local immigration events [11]. Potential causes of this increase in immigration include climate change-driven range shifts [12] and human introductions [13]. Although most immigrant species fail to establish large local populations [14], their presence in low numbers may increase the number of rare species within assemblages if the number of immigration events exceeds the number of local extinctions.

In this study, we ask whether there has been a detectable increase in numbers of locally rare species within assemblages across the globe. Also, we seek to shed light on the processes driving any detected increases in rare taxa. Quantifying the relationship between different facets of local biodiversity change helps us distinguish between alternative processes of change. While both processes introduced above, (1) *decreasing population sizes* and (2) *increasing immigration,* may increase the number of rare species locally, they will have different consequences for species richness and assemblage size (measured in terms of numbers of individuals within an assemblage). While these two processes are not mutually exclusive, in practice one may dominate over the other. We can determine their relative prevalence by looking at the relationship between trends in rarity, species richness and assemblage size. If process (1) *decreasing population sizes* dominates, we foresee two potential outcomes. In both outcomes, rarity is caused by a decline in population sizes of species already present. In the first of the two outcomes, however, there is no corresponding increase in net local extinctions and so we expect no change in species richness, a decline in assemblage size, and no relationship between changes in species richness and assemblage size (figure 1.1*a*). The alternative outcome for process (1) *decreasing populations* will arise if there is an increase in net extinction with increasing rarity. In this case, we expect covarying negative trends in species richness and assemblage size (figure 1.1*b*). In both outcomes, assemblage size should decline with increasing rarity due to fewer individuals being present in the assemblage.

In process (2) *increasing immigration*, where increasing rarity is mainly driven by an increase in net immigration, we expect trends in rarity, species richness and assemblage size to all be positive and covary [13] (figure 1.2). It is worth noting that increased immigration of species could cause no directional relationship between species richness and rarity like that predicted in figure 1.1*b* if increasing immigration of rare species balances increasing species losses through local extinction. The relationship between rarity and assemblage size would differ, though, as there would be no directional relationship between trends in these facets of biodiversity, nor a directional relationship between trends in assemblage size and species richness.

To further differentiate between the two processes potentially driving change in rarity, we then assess the net balance of immigration to local extinction among rare species. If process (1)—*decreasing populations*—is driving increasing rarity then we expect either a balance of extinction and immigrations among rare species (figure 1.1*a*) in line with the balance found within entire assemblages by Dornelas *et al.* [15], or more extinctions than immigrations among rare species (figure 1.1*b*). Conversely, if process (2)—*increasing immigration* (figure 1.2)—is driving increasing rarity then we expect there to be a higher rate of immigration events than extinctions among rare species.

# 2. Methods

For this analysis, we used a subset of 101 assemblage time series from the BioTIME database [16] that contained at least 10 years of monitoring data and numerical abundance count data (see electronic

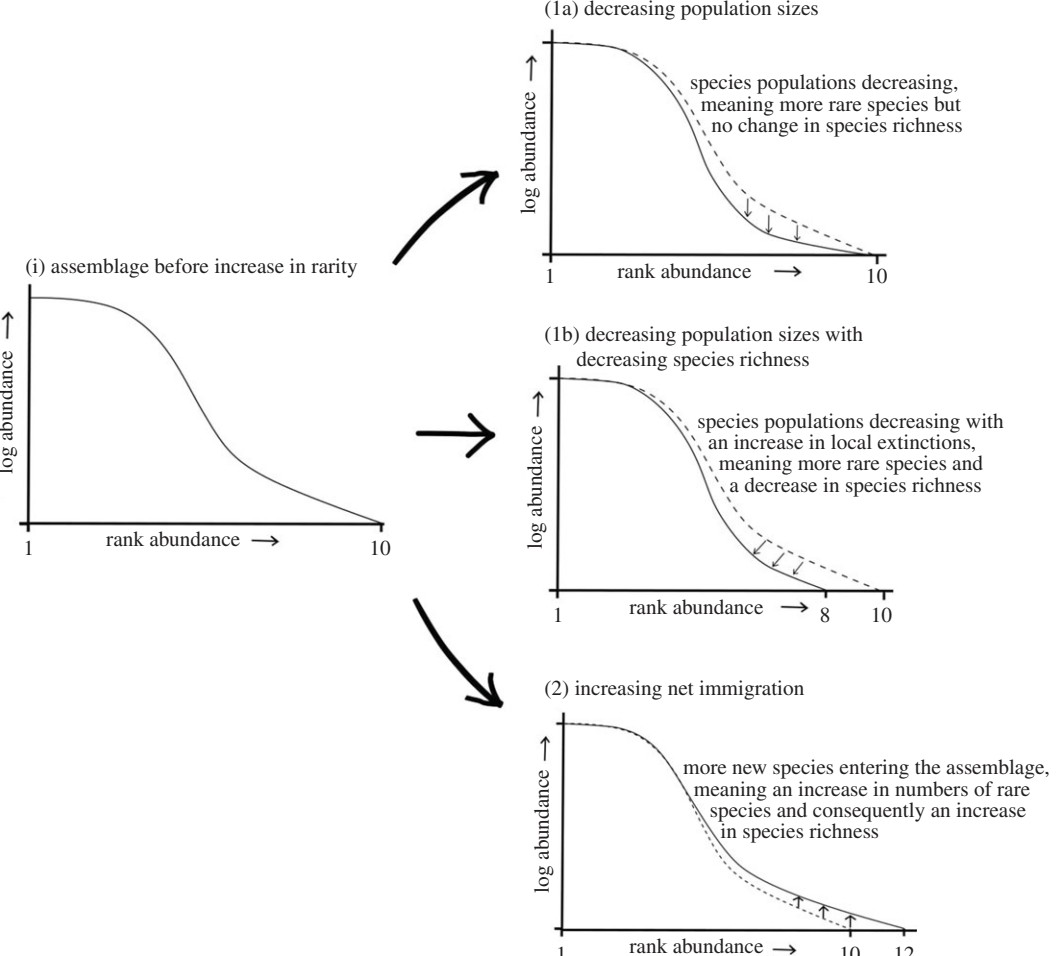

**Figure 1.** A conceptual framework of how shifts in numbers of rare species may relate to species richness change under a scenario of increasing rarity. Plot (i) shows the shape of the rank abundance curve of an assemblage containing 10 species before it undergoes an increase in numbers of rare species. Plot (1a) illustrates how the assemblage structure and size changes if decreasing populations without increasing local extinctions are driving increasing rarity. In this case, there are still 10 species present, but many species now have lower abundances leading to no change in species richness but a decrease in assemblage size. This shift in the rank abundance curve could also arise if there is an increase in local extinctions as a consequence of declining populations, but also a similar increase in immigration of rare species into the assemblage that prevents a net increase in extinction. Plot (1b) illustrates our expectation if decreasing populations are driving increasing rarity but with a net increase in local species extinctions due to population extinctions. In this case, there are now 8 rather than 10 species present in addition to lower species abundances among uncommon species. In this outcome, species richness and assemblage size should decrease as rarity increases. Plot (2) is the alternative process where increasing net immigration rather than decreasing populations is the main driver of increasing rarity. In our example, the original 10 species count has now risen to 12 species because of the addition of two new rare species. A small increase in assemblage size should also occur here because the newly arrived species have very low population sizes.

supplementary material, table S1 and figure S1 for full list). The BioTIME database consists of assemblage time series in which sites have been monitored using a consistent methodology. Like all ecosystems on the planet, these assemblages are impacted by combinations of global stressors [17]. Specifically, BioTIME time series are affected by the ubiquitous climate change [18], marine data include locations affected by overfishing [19], and most of the range in forest loss found across the planet is covered by BioTIME [20]. The widespread compositional change detected within BioTIME assemblages is indicative of change captured by these data [21]. We note that our analysis applies only to the locations and periods covered by these data, but these data are useful for gaining the nuanced overview of biodiversity change needed to avoid drawing overly simplistic results from few local studies [22].

We analysed 42 marine, 49 terrestrial and 10 freshwater assemblages distributed across the globe. Taxa represented include plants, fish, birds, mammals and invertebrates. Before analysis, data

underwent sample-based rarefaction to account for variation in sampling effort (*sensu* Dornelas *et al.* [11]). For this rarefaction process, each assemblage dataset was rarefied separately. For each assemblage, years with fewer than half the mean number of sampling occasions were excluded from the analysis. The minimum number of samples in a year was then extracted, and this number of samples was randomly selected from each of the other years. We retained the integrity of the assemblage sample within each rarefaction iteration. See electronic supplementary material, figure S2 for a schematic workflow of the rarefaction process. To ensure our results were robust to the random samples selected by the rarefaction process we repeated the rarefaction process 20 times. For each iteration, we retested trends in rarity and species richness as well as the relationship between these trends. Only the results of the first iteration are presented below, but plots of the distribution of results across the 20 rarefaction iterations can be seen in electronic supplementary material, figures S3 (overall slopes of change) and S4 (comparisons of slopes of change for each assemblage). For faunal non-sessile studies with multiple samples per year, we also removed the effects of seasonality by summing the abundance of each species within each year. There were no floral or sessile faunal assemblages that had been resampled within years. Analysis was undertaken in R [23].

We defined rarity based on local abundances of species. A commonly used definition of local rarity is the number of singletons, i.e. the number of species represented by a single individual within an assemblage [24,25]. Singletons are detected in many assemblages even with a high sampling effort [26,27]. A less conservative but strongly linked definition of rarity is the number of singletons plus the number of doubletons, i.e. the number of species represented by two or more individuals. We tested how both metrics of rarity changed so we can be sure any effect detected is not sensitive to the definition used. We focus on the results of the singletons and doubletons combined rather than singletons throughout the analysis. To ensure our results were robust to the quantification method of local rarity, we also assessed change in Fisher's alpha [1]. Fisher's alpha is a parameter of the logseries species abundance distribution model and is usually approximately equal to the number of singletons [28]. Fisher's alpha was calculated using the *fisher.alpha* function of the R package *preseqR* [29,30]. We ran our analysis with all three metrics of rarity.

We asked whether there is a systematic increase in rarity, measured in terms of the number of singletons and doubletons, using a negative binomial mixed-effect model using the R package *brms* [31] and the default priors. For testing the sensitivity of our results to the rarefaction process, however, we used the same model structure in a non-Bayesian package *GLMMadaptive* [32]. Our results were consistent between the *brms* and *GLMMadaptive* models (electronic supplementary material, figure S5). For the rarity model, the number of singletons and doubletons each year was regressed against mean centred year, where each year was centred around the mean year for the appropriate assemblage time series. Assemblage ID was included as a random effect with varying slope and intercept to calculate individual assemblage rates of change, and a single overall slope was estimated for global rarity change. We ran an identical model for singletons only (using *GLMMadaptive*), and an equivalent model for Fisher's alpha. The Fisher's alpha model was constructed using the *lme* function of the R package *nlme* [33], and included a power variance covariate structure given by the fitted values of the fitted model. This variance structure accounted for the increased residual variance of assemblages with greater fitted values for Fisher's alpha.

To explore how rarity changed with species richness and assemblage size (defined as the number of individuals within as assemblage), we also fitted two models: a mixed model of $\log_{10}$ species richness change regressed against mean centred year and a mixed model of $\log_{10}$ assemblage size regressed against mean centred year. Both models included Assemblage ID as a random effect and a Gaussian error distribution. We fitted these models in *brms* for the main analysis and *nlme* for testing the effect of the rarefaction process. The results from our models were consistent between *brms* and *nlme* (electronic supplementary material, figures S6 and S7). The strength and direction of the relationship between rates of change of rarity, rates of change of species richness and rates of change of assemblage size for each time series was then assessed by extracting individual assemblage rates of change from the random effects of each model and assessing correlation using Pearson's correlation coefficient. To ensure our results were not being overly influenced by extreme values, we undertook sensitivity testing by running the correlation tests 200 times while randomly removing 5%, 10% and 20% of the assemblage time-series trends.

To directly test whether there were more rare species immigrating than going locally extinct within assemblages, we used a methodology similar to that of Dornelas *et al.* [15]. We first selected population data for the rare species in our dataset. Our selection criterion was population data of any species that had an abundance of one or two for at least one of the sampling years. This resulted in

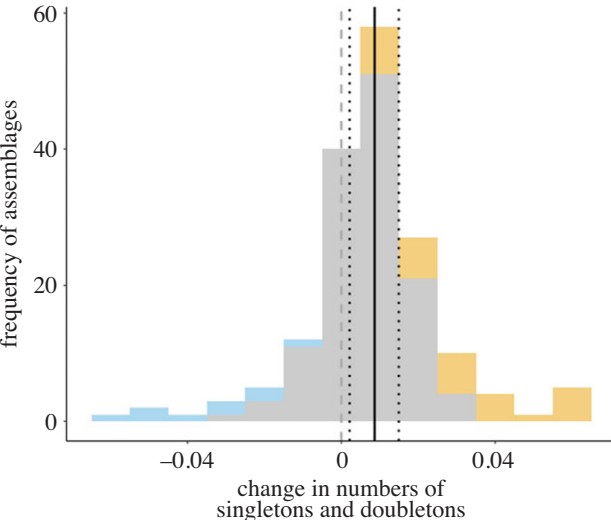

**Figure 2.** The distribution of mean slopes of change in the number of rare species within assemblages (defined as singletons and doubletons) calculated from the mixed model. Grey bars are where the 95% HPDI includes both positive and negative values, so the assemblage slope of change is not likely to be either positive or negative. The yellow bars are when the 95% HPDI of the slopes of change fall above 0, and the blue bars where the 95% HPDI of slopes of change fall below 0, so they represent assemblage slopes that are likely to be different from 0. The solid black line represents the mean overall global trend in changing numbers of rare species, and the dotted black lines represent the upper and lower 95% HPDI. The dashed grey line displays the 0 (no systematic trend) mark so that it is clear that the lower 95% HPDI of the main slope falls above 0 and so the model suggests a general increase in rarity.

selecting 14 635 populations. We retained the within-species population dynamics of each selected species population, meaning population abundances in some years could rise above or fall below one or two individuals. Some species were rare in more than one assemblage, and so accounted for multiple populations within the analysis. For each population, abundances were converted to a string of binary presence and absence data (1s and 0s). We then counted the number of immigrations (transition between 0 and 1) and extinctions (transition between 1 and 0). To avoid detection errors inflating immigration and extinction events, we applied the *runs.test* function from the *tseries* R package [34]. Species populations that only immigrated once into an assemblage were assigned to the category of 'Immigration', and species that went extinct only once were categorized as 'extinct'. Species that underwent more than one immigration or extinction event were categorized as 'multiple'. Any species with population abundances consistently above 0, and species without significant extinctions or immigrations, were categorized as 'persistent'. We compared the proportion of immigration to local extinction events among rare species to the proportion found by Dornelas *et al.* [15] for species of all abundance classes.

## 3. Results

We detected an increase in the number of rare species overall, defined in terms of singletons and doubletons (mean slope: 0.0087, lower 95% highest posterior density index (HPDI): 0.0022, upper 95% HPDI: 0.0149; figure 2*a*). The increase in rarity is less pronounced but still positive for rarity measured using numbers of singletons only, rather than singletons and doubletons (electronic supplementary material, figure S8). There was also a positive trend detected using Fisher's alpha (electronic supplementary material, figure S9), but this trend was not as strong. Changes in singletons correlate very closely to changes in singletons and doubletons ($r = 0.90$) and changes in Fisher's alpha correlated weakly with slopes of change of singletons and doubletons ($r = 0.20$).

We detected a positive trend in species richness (mean slope: 0.0025, lower 95% HPDI: 0.0007, upper 95% HPDI: 0.0043; electronic supplementary material, figure S10) and a less strong but still positive trend in assemblage size (mean slope: 0.003, lower 95% HPDI: −0.0015, upper 95% HPDI: 0.008; electronic supplementary material, figures S6 and S7). The positive

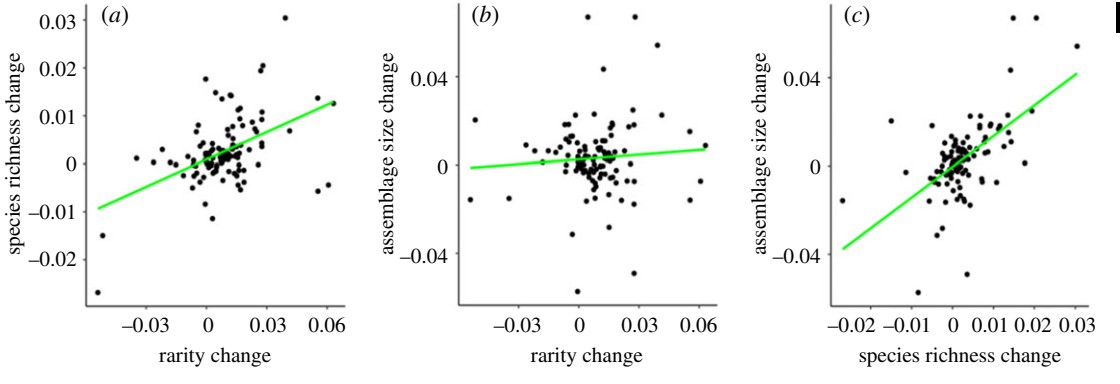

**Figure 3.** The relationship between the slopes of change of the number of rare species (singletons and doubletons) and species richness (*a*); the number of rare species and assemblage size (*b*); and species richness and assemblage size (*c*) within assemblages. The directions of the relationships are represented by linear trend lines. There are positive relationships between rarity and species richness change and assemblage size and species richness change, but no trend evident between rarity and assemblage size change.

trends detected in rarity, species richness and assemblage size were all robust to the variation introduced by the rarefaction process (electronic supplementary material, figure S3). There was a positive relationship between a change in the number of rare species and a change in species richness species within assemblages ($r = 0.48$; figure 3*a*), whereas no relationship was detected between change in rarity and change in assemblage size (0.07; figure 3*b*). A positive relationship was, however, detected between change in species richness and change in assemblage size (0.58; figure 3*c*). The positive relationship between rarity change and species richness change was also seen when measuring rarity as the number of singletons (electronic supplementary material, figure S12) and Fisher's alpha (electronic supplementary material, figure S13). The relationships between rarity, species richness and assemblage change were robust to sensitivity testing (electronic supplementary material, figure S14) and the rarefaction process (electronic supplementary material, figure S4).

Of the 14 635 rare species populations tested, 12 215 (83%) were persistent within assemblages. Of the rare species that were not persistent over time, 1618 (11%) underwent multiple immigration and extinction events, 518 (3.5%) immigrated into assemblages, and 284 (1.9%) went locally extinct.

## 4. Discussion

In line with our expectations, we found an increase in the number of rare species within assemblages, defined as the number of singletons and doubletons (figure 2). This result did not change when rarity was defined as the number of singletons (electronic supplementary material, figure S8), but it was less clear when evaluated using Fisher's alpha (electronic supplementary material, figure S9). The overall increase in the number of rare species within assemblages was correlated with increasing species richness (figure 3*a*), but not with increasing assemblage size (figure 3*b*). The positive relationship between change in rarity and change in species richness, coupled with the higher number of rare species immigrating than going locally extinct, is consistent with our predictions of process (2) *increasing immigration*, suggesting new species entering assemblages in low numbers are contributing to the maintenance of or increases in local species richness (figure 1.2). Our results, though, do not suggest that process (1) *decreasing populations* is not also contributing to increasing rarity, but that process (2) *increasing immigration* is the main driver of detected changes. Further analysis focusing on tracking individual species populations within assemblages is required to elucidate the proportion of species experiencing population declines versus immigration events.

The lack of a relationship between change in rarity and change in assemblage size was not in line with our predictions (figure 1) but may be explained by the small proportion of overall assemblage size that rare species populations account for. By definition, rare species have low abundances and hence contribute little to total assemblage abundance. Our results, therefore, imply a net increase in immigration. This scenario suggests that there are changes in the structure and identity of species present within assemblages and raises questions about the source and potential effect of increased immigration of rare species into assemblages.

We detected a trend of increasing immigration across over 100 assemblage studies from various taxa and realms, but how general are these findings? As noted above, locations sampled in BioTIME, and

consequently our analysis, include land- and seascapes affected by human drivers including overfishing, land-use intensification and climate change [18–21]. Our results will, therefore, be relevant to a substantial fraction of global assemblages and provide an instructive indication of how such assemblages are changing. However, we recognize that the assemblages in our analysis are not a random subset of all ecological systems so caution is needed in interpreting the findings. For example, BioTIME data is biased towards temperate regions, the Northern Hemisphere, and also taxonomically towards fish, birds and plants. The BioTIME database compiles data from systematically sampled assemblage time series, and thus can only include assemblages that have already been monitored. This also means BioTIME does not include before-after-control-impact studies, and it is likely that habitats undergoing radical transformations due to severe impacts deviate from the mean trends detected in our models. A truly representative biodiversity change dataset is a key challenge for the future but unfortunately not yet available [22]. The analysis described in this paper does, however, benchmark changes in rarity for those assemblages for which quality time-series data already exist.

Although the difference between the number of immigration and extinction events among rare species was not large (3.5% immigrations versus 1.9% extinctions), it is still a larger proportional difference than the 8% immigrations versus 7% extinctions detected by Dornelas $et\ al.$ [15] when they assessed immigration and local extinctions in entire assemblages. Although the proportional differences are small, they relate to large numbers of species (greater than 250 species). In addition, although the trend we detected in species richness across the 101 assemblages has a very high probability of being positive (95% HPDI between 0.0007 and 0.0043), our analysis used a $\log_{10}$ transformation. This means the estimated mean slope value translates to an increase on average of 1 species per year ($10^{0.0007}=1.002$, $10^{0.0043}=1.010$) which is in the same order of magnitude as the difference between the number of species immigrating and going extinct from our analysis of population trends. Increasing rare species immigration is also unlikely to be the only influence on species richness. For example, immigrant taxa do not necessarily always enter assemblages as rare species. Regardless, the potential influence of the immigrant rare species we detected will depend largely on whether their presence is due to insufficient time passing for many of the newly arrived species to have increased in abundance past singleton or doubleton status, or whether we are detecting an increase in transient species [35] that will not lead to increasing numbers of abundant immigrants but instead to a 'transient biodiversity surplus' [36].

One previously unrecognized facet of biodiversity change is that abundant species are maybe being replaced by rarer ones. This could contribute to reductions in overall assemblage abundance, for example as reported in [37,38]. However, we found no evidence of assemblage size decreasing on average, nor of assemblage size decreasing with increasing rarity. That assemblage size generally increased with increasing species richness but showed no trend with rarity suggests that processes other than increased immigration (figure 3c) are also influencing species richness.

It is clear from our results that there are substantial changes within assemblages, which in turn raises important questions around why there are more species immigrations among rare species. Human-mediated immigrants, often termed 'introduced species', are one possible source of rare species. The success of some species in spreading across the world through human vectors is such that we should now perhaps consider increasing local species richness as the null expectation [13].

Another possible driver of increased immigration could be migrations associated with climate change because changes in the suitability of habitats can force species to colonize new regions [36]. Many species are shifting poleward as a response to increasing temperatures [12,39]. As with many collations of ecological data, we had an overrepresentation of northern temporal ecosystems within our analysis (electronic supplementary material, figure S1). Increasing species richness is linked to warming temperatures in the temperate regions [18], so if climate migrants are driving increasing rarity and species richness patterns then the situation in tropical regions may be quite different.

A further issue in assessing biodiversity change within assemblages is that we never have complete samples. Estimations of sample completeness must take place so the 'true' number of species present in the assemblage can be deduced, and the ratio of singletons and doubletons within samples is often used for this purpose [40–42]. In our analysis, however, we found evidence of shifting numbers of singletons and doubletons within assemblages. The assemblage datasets included in the BioTIME database draw on consistent monitoring protocols, and rarefaction was applied to allow for meaningful comparisons of assemblage structure. Our results, therefore, suggest that the shifts in the presence of rare species are a consequence of genuine shifts in assemblage structure rather than sampling incompleteness. As such, relying on ratios of rare species to judge sample incompleteness requires caution because shifts in numbers of rare species could be erroneously attributed to changes in assemblage species richness.

In conclusion, we found evidence of increasing numbers of rare species entering assemblages. Biotic homogenization, where a few species win at the cost of many losers [43,44], is a possible outcome of this increased immigration. In addition, species that are lost may differ in functional traits from incoming species, therefore extensive non-native introductions may cause shifting ecosystem functioning [13]. Where these species are coming from, and how their presence may affect ecosystem functioning, becomes the next question. If many of the species are not establishing in their new assemblages, though, then a transient biodiversity surplus may mask important declines in resident assemblages.

Ethics. The BioTIME project (250189) received ethical clearance from the ERC in 2009, and the University of St Andrews, School of Biology, bioethics committee approved continued analyses of BioTIME data in April 2019.

Data accessibility. Data are available in [16]. Full details in electronic supplementary material, table S1.

Authors' contributions. F.A.M.J. undertook analysis, M.D. and A.E.M. supervised analysis and all authors wrote the manuscript.

Competing interests. We declare we have no competing interests.

Funding. F.A.M.J.'s PhD was financed by the School of Biology, University of St Andrews. M.D. and A.E.M. acknowledge funding by the Leverhulme Trust. A.E.M. acknowledges funding from the European Research Council (ERC AdG BioTIME 250189 and ERC PoC BioCHANGE 727440).

Acknowledgements. We thank the three anonymous reviewers for their constructive and insightful comments throughout the publication process.

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
