## [Reviewer comments · Royal Society Open Science]

Review History

RSOS-192045.R0 (Original submission)

Review form: Reviewer 1

Is the manuscript scientifically sound in its present form?

No

Are the interpretations and conclusions justified by the results?

No

Is the language acceptable?

Yes

Do you have any ethical concerns with this paper?

No

Have you any concerns about statistical analyses in this paper?

No

Recommendation?

Reject

Comments to the Author(s)

The key question posed by this manuscript, whether rarity in species assemblages is changing through time and if so how, is a perfectly reasonable and potentially important one. However, the answer will depend almost entirely on the study assemblages that are chosen. If a general answer is being sought, then these assemblages will need to be representative of those at large. The manuscript is framed as providing such a general answer. However, it is virtually silent on the issue of how representative the assemblages it is based on actually are.

This is not a trivial issue, because a vociferous debate has centred on the adequacy of the database that is used in the manuscript for providing general answers to questions of the dynamics of biodiversity. There seems good reason to believe that this database does not provide a representative sample of assemblages, although it is extremely difficult if not impossible categorically to demonstrate whether this is or is not the case. Given this it is difficult to understand how, as reflected for example in the abstract, the manuscript can make such statements as 'Our results are clear evidence of measurable changes in the structure of assemblages in the recent past'. For these assemblages that may be the case, but there is no evidence provided that this is true of assemblages in general.

At the very least, the text throughout this manuscript needs to be heavily revised to be clear that the generality of the findings may be very limited.

In addition to this rather fundamental problem, the manuscript raises the following concerns:

(i) The underlying theoretical model (Fig. 1) assumes that under decreasing population sizes, all species nonetheless remain in an assemblage. But surely, under such a scenario the most likely outcome is that at least some of these species go extinct, resulting in a decline in species richness? Note that the arrows in Fig. 1b are a bit misleading, as what is happening is a vertical movement from the dotted line down to the solid one, not a horizontal movement that seems to imply that species are increasing in their rank abundance (and likewise the opposite in Fig. 1c).

(ii) The methods state that 'Before analysis, data underwent sample based rarefaction to account for variation in sampling effort' (line 67-68). It is important to have rather more information here, without having to go to the paper referenced. Depending on just how it is done, such rarefaction could have different effects on the subsequent analyses, and certainly seems unlikely to have no effect. Indeed, given that many rarefaction approaches have a stochastic component to them when it comes to effects on individual species abundances, it is unclear why this is not accounted for at all in the analyses reported. This may be particularly critical given that the levels of immigration and extinction among rare species are small.

(iii) The focus of the analyses is on how the number of rare species changes with species richness, the assumption being that the former is a key determinant of the latter. However, an alternative model is that the overall abundance of individuals in an assemblage is a much more important determinant of its species richness. Is this a weaker predictor of species richness than is the number of rare species?

(iv) The documented relationship between change in number of rare species and change in species richness (Fig. 3) seems very dependent on a small number of outliers. Some kind of sensitivity testing would seem in order.

(v) Finally, despite its bold claims, as to how rarity is changing within assemblages, the overall levels of extinction and immigration are small (1.9% and 3.5%, respectively). They are thus clearly not a major determinant of the compositional structure of these assemblages. Indeed, at these kinds of levels many might simply have dismissed the findings as rather uninteresting.

Review form: Reviewer 2

Is the manuscript scientifically sound in its present form?

Yes

Are the interpretations and conclusions justified by the results?

Yes

Is the language acceptable?

Yes

Do you have any ethical concerns with this paper?

No

Have you any concerns about statistical analyses in this paper?

No

Recommendation?

Accept with minor revision (please list in comments)

Comments to the Author(s)

Several recent global studies found no consistent decreasing trends in local diversity, but relatively strong shifts in composition. Studies that look at how communities are changing are therefore key to understand such temporal patterns in species assemblages. This study focuses on how rare species contribute to the patterns of change, testing two main predictions (conceptualised in Fig. 1). The data were time series derived from the BioTIME database. I have one major comment and several smaller questions and suggestions, which generally focus on clarifying particular terms or quantitative approaches.

My main comment is that when going from pattern to process, one should carefully consider all possibilities that could create a certain pattern. On line 51 in the introduction, the authors say that “If declines in population sizes explain an increase in rarity, there should be no corresponding increase in species richness (Figure 1B)”. One possibility that is missing at this point, I believe, is a no net change in richness that is driven by population declines leading to local extinction and counterbalanced by immigration, that is, species turnover. The next paragraph seems to suggest this possibility, but I was a bit confused here because it is unclear whether Fig. 1B only represents declines in abundance without extinction (cf. figure caption) or also species turnover (as suggested on lines 56-58). So, it would help readers a lot if the text clarifies and justifies why the focus is on the scenario of species decreasing in abundance without going extinct only (see also abstract). The text should also be consistent in this context.

Line 44: “two mechanisms” - I would rather use the term 'process' for the population declines and immigrations. The mechanisms driving these two processes can be very different and dependent on the species or systems where you look at.

Line 69: “summing the abundance of each species” - Doesn't it depend on the type of sampling whether or not taking a sum makes sense to remove the effect of seasonality? I'm working with plants and would never sum abundances when multiple records are available, because the same individuals are just counted multiple times then (especially for long-lived species that are just standing there). I can imagine that for many fauna communities, where individuals are actually captured and removed from the community, summing across sample occasions makes sense. Another point that should be clarified here as well, I think, is whether there are time series where the number of samples per year (if there are multiple) changes across time (e.g., more frequent

within-year sampling in the most recent years). If so, this could also influence abundances when taking a sum across samples.

Line 86: “mean centred year” - Year was centred within or across time series? I assume within, so that intercepts represent the number of singletons and doubletons for the central year in a series.

Line 88: the slope for global rarity change is not “calculated”, it is just the slope parameter of the year-effect in the statistical model? For the random slopes, you could say there are calculated, because in a model fitted with, for instance, lme they are not fitted as parameters.

Line 95: “strength and direction of the relationship between rates of change of rarity and rates of change of species richness for each study was then assessed” - So this is a correlation between the assemblage-level (i.e. per time series) slopes derived from the random effects of the previous mixed models? Please clarify. Also note that this analysis then ignores the uncertainty in these slopes, that is, time series in which the relationship with year comes with high uncertainty (std. error on the slope) are given equal weight as those in which the relationship is very tight. I know that models fitted with lme or similar functions cannot provide these std. errors and that it would require, for instance, MCMC methods to do so. But maybe the authors can simply acknowledge that the analysis does not account for uncertainty in the slopes, just to be clear about it. Essentially, this analysis is in fact a meta-analysis in which trends derived from independent assemblage time series data sets are compared, so readers may expect the dataset-level uncertainty to be accounted for.

Line 96: “study” - better to be consistent and refer to an assemblage or time series?

Line 160: “biotic homogenization is a possible outcome of the increased immigration” - Only if it is a relatively small set of species that is immigrating across many sites. If the immigrants are a diverse set of species, each of which only immigrate in few sites, there will be no homogenization. Do the authors have evidence from their analyses for patterns of homogenization?

Line 162: “may differ in functional traits” - As with the homogenization, this is largely unknown from the results of this study, right? Together with the previous comment, I guess that future work should mainly focus on the identity and functional attributes of the species that immigrate and replace those that go locally extinct. [Note after reading the last paragraph: this aspect is actually covered in the conclusions; maybe integrate the lines 160-164 in that concluding paragraph?]

Decision letter (RSOS-192045.R0)

17-Jan-2020

Dear Professor Jones,

The editors assigned to your paper ("Recent increases in assemblage rarity are linked to increasing local immigration") have now received comments from reviewers. We would like you to revise your paper in accordance with the referee and Associate Editor suggestions which can be found below (not including confidential reports to the Editor). Please note this decision does not guarantee eventual acceptance.

Please submit a copy of your revised paper before 09-Feb-2020. Please note that the revision deadline will expire at 00.00am on this date. If we do not hear from you within this time then it will be assumed that the paper has been withdrawn. In exceptional circumstances, extensions may be possible if agreed with the Editorial Office in advance. We do not allow multiple rounds

of revision so we urge you to make every effort to fully address all of the comments at this stage. If deemed necessary by the Editors, your manuscript will be sent back to one or more of the original reviewers for assessment. If the original reviewers are not available, we may invite new reviewers.

- Data accessibility

<http://datadryad.org/submit?journalID=RSOS&manu=RSOS-192045>

- Competing interests

- Authors' contributions

- Acknowledgements

- Funding statement

on behalf of Professor Michael Bruford (Associate Editor) and Kevin Padian (Subject Editor)
openscience@royalsociety.org

Associate Editor's comments (Professor Michael Bruford):

Your paper has received two thorough reviews with somewhat divergent opinions. I am willing to see a major revision of your manuscript but this will need to take account of the very serious concerns of referee 1, and will require re-review.

Editor comments:

Thanks for your submission. As you will see our two reviewers have some substantial concerns and are split on the suitability for eventual publication. If you choose to revise and resubmit, please consider these thoughts very carefully and specifically. If you need more time, advise the editorial office. It may be prudent not to overstate your conclusions, given that your sample of assemblages is limited, although you may have a good diversity of samples and the results may be consistent. Best wishes.

Comments to Author:

Reviewers' Comments to Author:

Reviewer: 1

Comments to the Author(s)

The key question posed by this manuscript, whether rarity in species assemblages is changing through time and if so how, is a perfectly reasonable and potentially important one. However, the answer will depend almost entirely on the study assemblages that are chosen. If a general answer is being sought, then these assemblages will need to be representative of those at large. The manuscript is framed as providing such a general answer. However, it is virtually silent on the issue of how representative the assemblages it is based on actually are.

This is not a trivial issue, because a vociferous debate has centred on the adequacy of the

database that is used in the manuscript for providing general answers to questions of the dynamics of biodiversity. There seems good reason to believe that this database does not provide a representative sample of assemblages, although it is extremely difficult if not impossible categorically to demonstrate whether this is or is not the case. Given this it is difficult to understand how, as reflected for example in the abstract, the manuscript can make such statements as 'Our results are clear evidence of measurable changes in the structure of assemblages in the recent past'. For these assemblages that may be the case, but there is no evidence provided that this is true of assemblages in general.

At the very least, the text throughout this manuscript needs to be heavily revised to be clear that the generality of the findings may be very limited.

In addition to this rather fundamental problem, the manuscript raises the following concerns:

(i) The underlying theoretical model (Fig. 1) assumes that under decreasing population sizes, all species nonetheless remain in an assemblage. But surely, under such a scenario the most likely outcome is that at least some of these species go extinct, resulting in a decline in species richness? Note that the arrows in Fig. 1b are a bit misleading, as what is happening is a vertical movement from the dotted line down to the solid one, not a horizontal movement that seems to imply that species are increasing in their rank abundance (and likewise the opposite in Fig. 1c).

(ii) The methods state that 'Before analysis, data underwent sample based rarefaction to account for variation in sampling effort' (line 67-68). It is important to have rather more information here, without having to go to the paper referenced. Depending on just how it is done, such rarefaction could have different effects on the subsequent analyses, and certainly seems unlikely to have no effect. Indeed, given that many rarefaction approaches have a stochastic component to them when it comes to effects on individual species abundances, it is unclear why this is not accounted for at all in the analyses reported. This may be particularly critical given that the levels of immigration and extinction among rare species are small.

(iii) The focus of the analyses is on how the number of rare species changes with species richness, the assumption being that the former is a key determinant of the latter. However, an alternative model is that the overall abundance of individuals in an assemblage is a much more important determinant of its species richness. Is this a weaker predictor of species richness than is the number of rare species?

(iv) The documented relationship between change in number of rare species and change in species richness (Fig. 3) seems very dependent on a small number of outliers. Some kind of sensitivity testing would seem in order.

(v) Finally, despite its bold claims, as to how rarity is changing within assemblages, the overall levels of extinction and immigration are small (1.9% and 3.5%, respectively). They are thus clearly not a major determinant of the compositional structure of these assemblages. Indeed, at these kinds of levels many might simply have dismissed the findings as rather uninteresting.

Reviewer: 2

Comments to the Author(s)

Several recent global studies found no consistent decreasing trends in local diversity, but relatively strong shifts in composition. Studies that look at how communities are changing are therefore key to understand such temporal patterns in species assemblages. This study focuses on how rare species contribute to the patterns of change, testing two main predictions (conceptualised in Fig. 1). The data were time series derived from the BioTIME database. I have one major comment and several smaller questions and suggestions, which generally focus on clarifying particular terms or quantitative approaches.

My main comment is that when going from pattern to process, one should carefully consider all possibilities that could create a certain pattern. On line 51 in the introduction, the authors say that “If declines in population sizes explain an increase in rarity, there should be no corresponding increase in species richness (Figure 1B)”. One possibility that is missing at this point, I believe, is a no net change in richness that is driven by population declines leading to local extinction and counterbalanced by immigration, that is, species turnover. The next paragraph seems to suggest this possibility, but I was a bit confused here because it is unclear whether Fig. 1B only represents declines in abundance without extinction (cf. figure caption) or also species turnover (as suggested on lines 56-58). So, it would help readers a lot if the text clarifies and justifies why the focus is on the scenario of species decreasing in abundance without going extinct only (see also abstract). The text should also be consistent in this context.

Line 44: “two mechanisms” - I would rather use the term 'process' for the population declines and immigrations. The mechanisms driving these two processes can be very different and dependent on the species or systems where you look at.

Line 69: “summing the abundance of each species” - Doesn't it depend on the type of sampling whether or not taking a sum makes sense to remove the effect of seasonality? I'm working with plants and would never sum abundances when multiple records are available, because the same individuals are just counted multiple times then (especially for long-lived species that are just standing there). I can imagine that for many fauna communities, where individuals are actually captured and removed from the community, summing across sample occasions makes sense. Another point that should be clarified here as well, I think, is whether there are time series where the number of samples per year (if there are multiple) changes across time (e.g., more frequent within-year sampling in the most recent years). If so, this could also influence abundances when taking a sum across samples.

Line 86: “mean centred year” - Year was centred within or across time series? I assume within, so that intercepts represent the number of singletons and doubletons for the central year in a series.

Line 88: the slope for global rarity change is not “calculated”, it is just the slope parameter of the year-effect in the statistical model? For the random slopes, you could say there are calculated, because in a model fitted with, for instance, lme they are not fitted as parameters.

Line 95: “strength and direction of the relationship between rates of change of rarity and rates of change of species richness for each study was then assessed” - So this is a correlation between the assemblage-level (i.e. per time series) slopes derived from the random effects of the previous mixed models? Please clarify. Also note that this analysis then ignores the uncertainty in these slopes, that is, time series in which the relationship with year comes with high uncertainty (std. error on the slope) are given equal weight as those in which the relationship is very tight. I know that models fitted with lme or similar functions cannot provide these std. errors and that it would require, for instance, MCMC methods to do so. But maybe the authors can simply acknowledge that the analysis does not account for uncertainty in the slopes, just to be clear about it. Essentially, this analysis is in fact a meta-analysis in which trends derived from independent assemblage time series data sets are compared, so readers may expect the dataset-level uncertainty to be accounted for.

Line 96: “study” - better to be consistent and refer to an assemblage or time series?

Line 160: “biotic homogenization is a possible outcome of the increased immigration” - Only if it is a relatively small set of species that is immigrating across many sites. If the immigrants are a diverse set of species, each of which only immigrate in few sites, there will be no homogenization. Do the authors have evidence from their analyses for patterns of homogenization?

Line 162: “may differ in functional traits” - As with the homogenization, this is largely unknown

from the results of this study, right? Together with the previous comment, I guess that future work should mainly focus on the identity and functional attributes of the species that immigrate and replace those that go locally extinct. [Note after reading the last paragraph: this aspect is actually covered in the conclusions; maybe integrate the lines 160-164 in that concluding paragraph?]

Author's Response to Decision Letter for (RSOS-192045.R0)

See Appendix A.

RSOS-192045.R1 (Revision)

Review form: Reviewer 2

Is the manuscript scientifically sound in its present form?

Yes

Are the interpretations and conclusions justified by the results?

Yes

Is the language acceptable?

Yes

Do you have any ethical concerns with this paper?

No

Have you any concerns about statistical analyses in this paper?

No

Recommendation?

Accept with minor revision (please list in comments)

Comments to the Author(s)

I have now carefully read the responses to my previous comments (referee 2) and am generally pleased with the way the authors handled them in preparing a revised manuscript. The authors' efforts to revise the conceptual diagram and to explain the possible scenario's more extensively throughout the text, will make the story much clearer for readers. There were some smaller points that I noticed when reading the manuscript again (I refer to line numbers of the manuscript version with track changes):

- Line 70-72 and Fig. 1A: "...if declines in population sizes without increases in local extinctions explain an increase in rarity, there should be no corresponding increase in species richness" Why linking local extinctions to increases in richness? It would be more logical if you say that there should be no changes in species richness (or no decreases in species richness, if you want to stress a direction). Also consider changing this in the text within Fig. 1A.
- Line 85: "...a balance of extinctions and colonisations..." In the rest of the text the term immigration is used. Better to be consistent (and check throughout the entire manuscript)?
- Line 222: typo? "We detect increases a trend in increasing immigration..."

- Line 244: two typo's in one sentence? "...facet of biodiversity change in that abundant species may be being replaced by rarer ones."

In addition to these smaller points, I also noticed that referee 1 commented about the representativeness of the database for biodiversity change across the globe. Yes, the data are obviously not representing sites where wholesale habitat conversion took place, but to me this fact doesn't take away any relevance of the study for the posed questions. Temporal data are just not available for many of these heavily impacted sites and, maybe more importantly, it is just not relevant to look at changes in population sizes and how they scale to diversity changes in case a system is completely converted (virtually all species are lost and replaced by a complete set of other taxa). Besides describing more clearly what type of sites your study is focusing on (as you did in response to the comment of referee 1), I think it is also important to argue why the data are relevant for studying biodiversity change. Besides the arguments I mentioned, there are probably more ways to justify the use of this database for this particular study. Being clear about it should avoid strong criticism afterwards.

Review form: Reviewer 3 (Cristina Banks-Leite)

Is the manuscript scientifically sound in its present form?

Yes

Are the interpretations and conclusions justified by the results?

Yes

Is the language acceptable?

Yes

Do you have any ethical concerns with this paper?

No

Have you any concerns about statistical analyses in this paper?

Yes

Recommendation?

Major revision is needed (please make suggestions in comments)

Comments to the Author(s)

I have received this manuscript for review after it having gone through one round of review. I have read the comments from the previous reviewers and believe that the issues they raised were appropriately dealt with, although I'll leave it to them and to the editor to make this decision. This is an interesting study which uses a very large dataset composed of 101 time-series to understand if patterns of rarity are changing over time, and if so why would that be the case. I enjoyed reading this, although I found a very large number of typos and awkward sentences that need to be thoroughly revised.

The explanation of species selection is confusing. In lines 107-118, it says that rare species were singletons and doubletons found within each year. Fine. Then in lines 147-149, the criterion for selecting rare species becomes "having a minimum population abundance of 1 or 2 within the assemblage", so does this mean all species >1 individual, including the most abundant ones? So then you're excluding the rare species which are the focus of this ms, right? Now, assuming that selection criterion for rare species here is the same as lines 107-118, in which rare species are the singletons and doubletons. If you select only the rare species for this analysis, and ignore information of abundant species, does this mean that some colonisations and some extinctions may have been abundant species that became rare or rare species that became abundant (i.e., they

were in and out of your selected dataset but not of the full dataset)? As you can see I'm very confused.

Of the issues raised by the previous reviewer, I would still like to highlight one. There were 518 cases of immigrations and 284 cases of extinction, so overall 234 new species over 101 assemblages. Of the populations tested, there were $(3.5-1.9)/1.6\%$ new species. I'm struggling to connect how such a small increase in number of species per timeseries/assemblage can lead to such strong increases in species richness. Assuming that these assemblages would have dozens of species, it's hard to see how one extra species would lead to a strong increase in richness over time. The authors included a new part of discussion to explain this, but I confess that it didn't fully convince me, and this may be in part because I didn't understand the immigration vs extinction analyses (comment above). Or maybe what needs to be done is to reassess whether the increases in richness are in fact strong, or just barely significant (as lower 95%HPDI is 0.0007) . Line 166 - here it says that the increase in rarity was more pronounced then using Fisher's alpha, but that's not what Fig S9 shows. SE overlaps 0.

Fig 3 - inclusion of panel C to respond to Reviewer 1 seems out of context here. I would suggest including this in the hypotheses, or as part of the trends you expect to find, because otherwise this will make little sense for further readers.

Decision letter (RSOS-192045.R1)

Dear Professor Jones:

On behalf of the Editors, I am pleased to inform you that your Manuscript RSOS-192045.R1 entitled "Recent increases in assemblage rarity are linked to increasing local immigration" has been accepted for publication in Royal Society Open Science subject to minor revision in accordance with the referee suggestions. Please find the referees' comments at the end of this email.

The reviewers and Subject Editor have recommended publication, but also suggest some minor revisions to your manuscript. Therefore, I invite you to respond to the comments and revise your manuscript.

- Ethics statement

- Data accessibility

If you wish to submit your supporting data or code to Dryad (<http://datadryad.org/>), or modify your current submission to dryad, please use the following link:
<http://datadryad.org/submit?journalID=RSOS&manu=RSOS-192045.R1>

- **Competing interests**

- **Authors' contributions**

- **Acknowledgements**

- **Funding statement**

Because the schedule for publication is very tight, it is a condition of publication that you submit the revised version of your manuscript before 02-May-2020. Please note that the revision deadline will expire at 00.00am on this date. If you do not think you will be able to meet this date please let me know immediately.

on behalf of Professor Michael Bruford (Associate Editor) and Kevin Padian (Subject Editor)
openscience@royalsociety.org

Associate Editor Comments to Author (Professor Michael Bruford):

We have received two reviews of your revision, one from an original reviewer, who is broadly satisfied with your revision and one from a new reviewer (sorry, but getting reviewers to re-review at this difficult time has been challenging) who raises an additional substantive point which I think needs to be addressed. Once these issues have been satisfactorily dealt with, I will recommend acceptance of this interesting study.

Reviewer comments to Author:
Reviewer: 2

Comments to the Author(s)

I have now carefully read the responses to my previous comments (referee 2) and am generally pleased with the way the authors handled them in preparing a revised manuscript. The authors' efforts to revise the conceptual diagram and to explain the possible scenario's more extensively throughout the text, will make the story much clearer for readers. There were some smaller

points that I noticed when reading the manuscript again (I refer to line numbers of the manuscript version with track changes):

- Line 70-72 and Fig. 1A: "...if declines in population sizes without increases in local extinctions explain an increase in rarity, there should be no corresponding increase in species richness" Why linking local extinctions to increases in richness? It would be more logical if you say that there should be no changes in species richness (or no decreases in species richness, if you want to stress a direction). Also consider changing this in the text within Fig. 1A.
- Line 85: "...a balance of extinctions and colonisations..." In the rest of the text the term immigration is used. Better to be consistent (and check throughout the entire manuscript)?
- Line 222: typo? "We detect increases a trend in increasing immigration..."
- Line 244: two typo's in one sentence? "...facet of biodiversity change in that abundant species may be being replaced by rarer ones."

In addition to these smaller points, I also noticed that referee 1 commented about the representativeness of the database for biodiversity change across the globe. Yes, the data are obviously not representing sites where wholesale habitat conversion took place, but to me this fact doesn't take away any relevance of the study for the posed questions. Temporal data are just not available for many of these heavily impacted sites and, maybe more importantly, it is just not relevant to look at changes in population sizes and how they scale to diversity changes in case a system is completely converted (virtually all species are lost and replaced by a complete set of other taxa). Besides describing more clearly what type of sites your study is focusing on (as you did in response to the comment of referee 1), I think it is also important to argue why the data are relevant for studying biodiversity change. Besides the arguments I mentioned, there are probably more ways to justify the use of this database for this particular study. Being clear about it should avoid strong criticism afterwards.

Reviewer: 3

Comments to the Author(s)

I have received this manuscript for review after it having gone through one round of review. I have read the comments from the previous reviewers and believe that the issues they raised were appropriately dealt with, although I'll leave it to them and to the editor to make this decision. This is an interesting study which uses a very large dataset composed of 101 time-series to understand if patterns of rarity are changing over time, and if so why would that be the case. I enjoyed reading this, although I found a very large number of typos and awkward sentences that need to be thoroughly revised.

The explanation of species selection is confusing. In lines 107-118, it says that rare species were singletons and doubletons found within each year. Fine. Then in lines 147-149, the criterion for selecting rare species becomes "having a minimum population abundance of 1 or 2 within the assemblage", so does this mean all species >1 individual, including the most abundant ones? So then you're excluding the rare species which are the focus of this ms, right? Now, assuming that selection criterion for rare species here is the same as lines 107-118, in which rare species are the singletons and doubletons. If you select only the rare species for this analysis, and ignore information of abundant species, does this mean that some colonisations and some extinctions may have been abundant species that became rare or rare species that became abundant (i.e., they were in and out of your selected dataset but not of the full dataset)? As you can see I'm very confused.

Of the issues raised by the previous reviewer, I would still like to highlight one. There were 518 cases of immigrations and 284 cases of extinction, so overall 234 new species over 101 assemblages. Of the populations tested, there were $(3.5-1.9)/1.6\%$ new species. I'm struggling to connect how such a small increase in number of species per timeseries/assemblage can lead to such strong increases in species richness. Assuming that these assemblages would have dozens of species, it's hard to see how one extra species would lead to a strong increase in richness over time. The authors included a new part of discussion to explain this, but I confess that it didn't

fully convince me, and this may be in part because I didn't understand the immigration vs extinction analyses (comment above). Or maybe what needs to be done is to reassess whether the increases in richness are in fact strong, or just barely significant (as lower 95%HPDI is 0.0007) . Line 166 – here it says that the increase in rarity was more pronounced then using Fisher's alpha, but that's not what Fig S9 shows. SE overlaps 0.

Fig 3 – inclusion of panel C to respond to Reviewer 1 seems out of context here. I would suggest including this in the hypotheses, or as part of the trends you expect to find, because otherwise this will make little sense for further readers.

Author's Response to Decision Letter for (RSOS-192045.R1)

See Appendix B.

RSOS-192045.R2 (Revision)

Review form: Reviewer 2

Is the manuscript scientifically sound in its present form?

Yes

Are the interpretations and conclusions justified by the results?

Yes

Is the language acceptable?

Yes

Do you have any ethical concerns with this paper?

No

Have you any concerns about statistical analyses in this paper?

No

Recommendation?

Accept as is

Comments to the Author(s)

The authors have carefully addressed my previous questions. Especially the updated sections in the methods and discussion on the representativeness and value of BioTIME for the research question of this study are a valuable addition to the manuscript. Note that I have only evaluated the changes that were made in response to my previous review and have not looked in detail at the many edits in the text that were made in response to reviewer 3.

Review form: Reviewer 3 (Cristina Banks-Leite)

Is the manuscript scientifically sound in its present form?

Yes

Are the interpretations and conclusions justified by the results?

Yes

Is the language acceptable?

Yes

Do you have any ethical concerns with this paper?

No

Have you any concerns about statistical analyses in this paper?

No

Recommendation?

Accept with minor revision (please list in comments)

Comments to the Author(s)

I appreciate the authors' responses to my comments. I am much happier with the text now. I do have just one question for the authors and would like them to consider whether including a sentence or two in the discussion would be justifiable. Do you think that tracking species identity for all these communities through time would have been feasible and would have resulted in a better estimation of whether the changes in rarity are caused by a mix of immigration and extinction? Because, as it's mentioned in the paper, the two processes are not mutually exclusive and my feeling from the methods is that they can't completely set apart the relative importance of extinction or population reductions and immigrations. Maybe I got this wrong and the authors are indeed doing that, in which case again adding a sentence about clarifying this would improve clarity.

Line 246 – change “may be being” to “are maybe being”

Decision letter (RSOS-192045.R2)

Dear Dr Jones:

On behalf of the Editors, I am pleased to inform you that your Manuscript RSOS-192045.R2 entitled "Recent increases in assemblage rarity are linked to increasing local immigration" has been accepted for publication in Royal Society Open Science subject to minor revision in accordance with the referee suggestions. Please find the referees' comments at the end of this email.

The reviewers and Subject Editor have recommended publication, but also suggest some minor revisions to your manuscript. Therefore, I invite you to respond to the comments and revise your manuscript.

- Ethics statement

- Data accessibility

<http://datadryad.org/submit?journalID=RSOS&manu=RSOS-192045.R2>

- Competing interests

- Authors' contributions

- Acknowledgements

- Funding statement

Because the schedule for publication is very tight, it is a condition of publication that you submit the revised version of your manuscript before 01-Jul-2020. Please note that the revision deadline will expire at 00.00am on this date. If you do not think you will be able to meet this date please let me know immediately.

To revise your manuscript, log into <https://mc.manuscriptcentral.com/rsos> and enter your Author Centre, where you will find your manuscript title listed under "Manuscripts with Decisions". Under "Actions," click on "Create a Revision." You will be unable to make your

revisions on the originally submitted version of the manuscript. Instead, revise your manuscript and upload a new version through your Author Centre.

on behalf of Professor Michael Bruford (Associate Editor) and Kevin Padian (Subject Editor)
openscience@royalsociety.org

Reviewer comments to Author:
Reviewer: 2

Comments to the Author(s)

The authors have carefully addressed my previous questions. Especially the updated sections in the methods and discussion on the representativeness and value of BioTIME for the research question of this study are a valuable addition to the manuscript. Note that I have only evaluated the changes that were made in response to my previous review and have not looked in detail at the many edits in the text that were made in response to reviewer 3.

Reviewer: 3

Comments to the Author(s)

I appreciate the authors' responses to my comments. I am much happier with the text now. I do have just one question for the authors and would like them to consider whether including a sentence or two in the discussion would be justifiable. Do you think that tracking species identity for all these communities through time would have been feasible and would have resulted in a better estimation of whether the changes in rarity are caused by a mix of immigration and extinction? Because, as it's mentioned in the paper, the two processes are not mutually exclusive and my feeling from the methods is that they can't completely set apart the relative importance of extinction or population reductions and immigrations. Maybe I got this wrong and the authors are indeed doing that, in which case again adding a sentence about clarifying this would improve clarity.

Line 246 – change “may be being” to “are maybe being”

Author's Response to Decision Letter for (RSOS-192045.R2)

See Appendix C.

Decision letter (RSOS-192045.R3)

Dear Dr Jones,

It is a pleasure to accept your manuscript entitled "Recent increases in assemblage rarity are linked to increasing local immigration" in its current form for publication in Royal Society Open Science.

on behalf of Professor Michael Bruford (Associate Editor) and Kevin Padian (Subject Editor)
openscience@royalsociety.org

Follow Royal Society Publishing on Twitter: [@RSocPublishing](https://twitter.com/RSocPublishing)
Follow Royal Society Publishing on Facebook:
<https://www.facebook.com/RoyalSocietyPublishing.FanPage/>
Read Royal Society Publishing's blog: <https://blogs.royalsociety.org/publishing/>

Appendix A

Editor comments:

Thanks for your submission. As you will see our two reviewers have some substantial concerns and are split on the suitability for eventual publication. If you choose to revise and resubmit, please consider these thoughts very carefully and specifically. If you need more time, advise the editorial office. It may be prudent not to overstate your conclusions, given that your sample of assemblages is limited, although you may have a good diversity of samples and the results may be consistent. Best wishes.

Comments to Author:

Reviewers' Comments to Author:

Reviewer: 1

Comments to the Author(s)

The key question posed by this manuscript, whether rarity in species assemblages is changing through time and if so how, is a perfectly reasonable and potentially important one. However, the answer will depend almost entirely on the study assemblages that are chosen. If a general answer is being sought, then these assemblages will need to be representative of those at large. The manuscript is framed as providing such a general answer. However, it is virtually silent on the issue of how representative the assemblages it is based on actually are.

This is not a trivial issue, because a vociferous debate has centred on the adequacy of the database that is used in the manuscript for providing general answers to questions of the dynamics of biodiversity. There seems good reason to believe that this database does not provide a representative sample of assemblages, although it is extremely difficult if not impossible categorically to demonstrate whether this is or is not the case. Given this it is difficult to understand how, as reflected for example in the abstract, the manuscript can make such statements as 'Our results are clear evidence of measurable changes in the structure of assemblages in the recent past'. For these assemblages that may be the case, but there is no evidence provided that this is true of assemblages in general.

At the very least, the text throughout this manuscript needs to be heavily revised to be clear that the generality of the findings may be very limited.

We thank the reviewer for their comment, and agree that we do not wish to overstate our results. To this end we have edited the text in various locations to address this issue. While we recognise that BioTIME data represents a biased and limited sample of the Earth's biodiversity, though, we disagree that our findings are very limited. As we mention in the sections added to the manuscript below, BioTIME is not a collection of pristine habitats. Anthropogenic impact on natural habitats are ubiquitous and complicated, and different systems are affected by different combinations of threats. BioTIME was designed as a collection of systematically sampled timeseries rather than before/after impact studies, but it contains data from heavily impacted areas. For example, the marine data includes assemblages from heavily fished locations, and some of the terrestrial assemblages are located in various ranges of forest loss. We acknowledge that BioTIME is spatially biased towards data in particular regions or for particular taxa, but this is a systematic problem across ecological knowledge. We know of no database that is available to conduct unbiased analysis of local biodiversity change, but this should not preclude efforts to understand biodiversity change.

We have added the below text to the manuscript to address the limitations of our results and make caveats clear.

Line 78-82: “The BioTIME database consists of assemblage time series in which sites have been monitored using consistent methodology. Like all systems on the planet, these assemblages are impacted by global stressors, particularly climate change, and many are also affected by local anthropogenic drivers such as overfishing or forest clearance. Nevertheless, we note that our analysis applies only to the locations and time spans covered by these data”

Lines 195-200: “We detect increases a trend in increasing immigration across over 100 studies across taxa and realms, but the BioTIME database was designed to collect data from systematically sampled assemblage time-series. This means BioTIME does not include before-after-control-impact studies, and it is likely that habitats undergoing radical transformations due to severe impacts deviate from the mean impacts in our models. However, the locations sampled in BioTIME and consequently our analysis include land- and seascapes affected by human drivers including overfishing, land use intensification and climate change.”

In addition to this rather fundamental problem, the manuscript raises the following concerns:

(i) The underlying theoretical model (Fig. 1) assumes that under decreasing population sizes, all species nonetheless remain in an assemblage. But surely, under such a scenario the most likely outcome is that at least some of these species go extinct, resulting in a decline in species richness? Note that the arrows in Fig. 1b are a bit misleading, as what is happening is a vertical movement from the dotted line down to the solid one, not a horizontal movement that seems to imply that species are increasing in their rank abundance (and likewise the opposite in Fig. 1c).

We agree with the reviewer that the potential effect of increasing local extinctions needed to be included in our conceptual framework. To this end we have added:

Line 37: “While rare species may be retained in the short term because of extinction debt, population declines could eventually lead to declining local species richness as rarity increases.”

Line 50-54: “In a situation where rarity is caused by a decline in population size of species already present, we would either expect no change in species richness or a decrease in species richness depending on whether numbers of local extinctions were also rising. In both cases we would expect a decline in assemblage size.”

Line 62: “Alternatively, Scenario B) illustrates our expectation if declining populations are coupled with increasing local extinctions (Figure 1B). Where increasing numbers of populations decline to local extinction, changes in species richness should negatively correlate with changes in rarity”.

Figure 1: we added a third scenario (Figure 1B/**Figure R1**) to our conceptual diagram to describe what we would expect with more local extinctions

We also updated **Figure 1/Figure R1** and so that the arrows reflect the horizontal movements implied by our concepts.

Figure R5, (Figure 1). A conceptual framework of how shifts in numbers of rare species may relate to species richness change under a scenario of increasing rarity. Plot (i) shows the shape of the rank abundance curve of an assemblage containing 10 species before it undergoes an increase in numbers of rare species. Plot (A) illustrates how the assemblage structure and size changes if decreasing populations without increasing local extinctions are driving increasing rarity. In this case there are still 10 species present, but many species now have lower abundances leading to no change in species richness but a decrease in assemblage size. This scenario could also arise if decreasing populations lead to increasing local extinctions, but a corresponding increase in immigration balances it. Plot (B) illustrates our expectation if there are net increases in local species extinctions due to population extinctions, where there are now 8 rather than 10 species present in addition to lower species abundances among uncommon species. In this case species richness and assemblage size should decrease as rarity increases. Plot (C) is the alternative relationship between species richness and increasing rarity, where increasing net immigration of rare species into the assemblage is the main driver of change in rarity. In our example, the original 10 species count has now risen to 12 species because of the addition of two new rare species leading to an increase in species richness. A small increase in assemblage size should also occur here because the newly arrived species have very low population sizes.

(ii) The methods state that 'Before analysis, data underwent sample based rarefaction to account for variation in sampling effort' (line 67-68). It is important to have rather more information here, without having to go to the paper referenced. Depending on just how it is done, such rarefaction could have different effects on the subsequent analyses, and certainly seems unlikely to have no

effect. Indeed, given that many rarefaction approaches have a stochastic component to them when it comes to effects on individual species abundances, it is unclear why this is not accounted for at all in the analyses reported. This may be particularly critical given that the levels of immigration and extinction among rare species are small.

We thank the reviewer for this suggestion, and on refection agree that some indication of the sensitivity of our results to the rarefaction process is needed. We therefore re-rarefied the data a further 19 times and then re-ran our analysis. We chose to use the frequentist rather than Bayesian modelling technique to accomplish this analysis because of computing restrictions. Our results were robust to the rarefaction variation, as we found overall positive trends for each of the 20 iterations of the rarefaction process (**Figure R2/****Figure S3** for overall trends; **Figure R3/****Figure S4** for covariances).

Figure R2. The distribution of estimated slopes of change in numbers of rare species in terms of singletons and doubletons, species richness and assemblage size over time, as estimated 20 times after rarefaction. The positive results of the models were robust the variation introduced by rarefaction.

Figure R3. The distribution of correlation values between slopes of change of rarity (singletons and doubletons), species richness and assemblage size across the results from the 20 different iterations of the rarefaction process. A positive relationship between rarity and species richness is evident in each iteration, so the result is robust to the variation introduced by the rarefaction process.

The details of rarefaction are also indeed not trivial, and we agree on further consideration that more detail is needed within the manuscript. We have therefore added additional text to the method section from line 93-103: “For this rarefaction process, each assemblage dataset was rarefied separately. For each assemblage, years with fewer than half the mean number of sampling occasions were excluded from the analysis. The minimum number of samples in a year was then extracted, and this number of samples was randomly selected from each of the other years. We retained the integrity of the assemblage sample within each rarefaction iteration. See **Figure S2(Figure R4)** for a schematic workflow of the rarefication process. To ensure our results were robust to the random samples selected by the rarefaction process we repeated the rarefaction process 20 times. For each iteration’s data we re-tested trends in rarity and species richness as well as the relationship between these trends. See below for more detail on analysis. Only the results of the first iteration are presented below, but plots of the distribution of results across the 20 rarefaction iterations can be seen in **Figure S3**(overall slopes of change) and **Figure S4** (comparisons of slopes of change for each assemblage).

Figure R4. An illustration of the rarefaction workflow applied to the biotime database before analysis. It is now included as Figure S1 of the supplementary information.

(iii) The focus of the analyses is on how the number of rare species changes with species richness, the assumption being that the former is a key determinant of the latter. However, an alternative model is that the overall abundance of individuals in an assemblage is a much more important determinant of its species richness. Is this a weaker predictor of species richness than is the number of rare species?

We thank the reviewer for their suggestion, and agree that overall abundance may also predict species richness. We chose to focus on numbers of rare species for this analysis because of the influence of rare species on species richness and rare species potential sensitivity to change, but have now included information on how species richness is changing with assemblage size. We found that species richness does indeed positively correlate with assemblage size, and that assemblage size is potentially increasing across the studies analysed. These results tie in with the suggestion from the rarity section of the analysis (that there is increased immigration) because it suggests that the new species entering assemblages are not simply replacing already present species populations. Contrary to our expectation from Figure 1C, however, numbers of rare species did not positively correlate with assemblage size. On reflection, this makes sense because singletons and doubletons by their nature will likely make up small percentages of the overall assemblage size. We do not think, though, that the influence of assemblage size on species richness invalidates the results or interest in increasing rarity. As we consider in our discussion, a signal of increasing numbers of species immigrating in small numbers is interesting despite their negligible effect on assemblage size because the effects of immigrating species may increase substantially over time as some species become established. In addition, species richness is still an important measurement for assessing biodiversity at this local scale, and as we show the presence of rare species influences species richness.

To address the reviewer's concerns, we have added an analysis of assemblage size and how change in assemblage size relates to species richness and rarity to our manuscript. We added predictions of how assemblage size should change according to our two processes in the introduction, methods, and figure legend. We include the results of how assemblage size is changing across our data in the supplementary material (**Figure S11**) and plots of how assemblage size changes with change in rarity

(**Figure 3B**) and species richness (**Figure 3C**), and the following text to the discussion: Lines 210-215 “we found no evidence of assemblage size decreasing on average, nor of assembled size decreasing with increasing rarity. That assemblage size generally increased with increasing species richness but showed no trend with rarity suggest that processes other than increased immigration (Figure 3C) are also influencing species richness.”

(iv) The documented relationship between change in number of rare species and change in species richness (Fig. 3) seems very dependent on a small number of outliers. Some kind of sensitivity testing would seem in order.

We acknowledge that the relationship does seem potentially driven by outliers, and welcome the suggestion of sensitivity testing to ensure the robustness of our results. To this end, we ran the correlation tests 200 times while randomly removing 5%, 10% and 20% of the assemblage time-series trends. The results of these bootstraps, shown in **Figure R5** for this review and in **Figure S14** of the supplementary material, suggest that our result of a positive relationship between species richness and rarity is robust to removal of data points. To clarify this issue for readers we have included the following text in Line 182: “The relationships between rarity, species richness and assemblage change were also robust to sensitivity testing (**Figure S14**)”.

Figure R5. Testing the sensitivity of the relationship between species richness change and rarity change (singletons and doubletons) within assemblages. We randomly removed a subset of 5% (panel a), 10% (panel b) and 25% (panel c) assemblages and then ran a Pearson's Correlation test. The uncertainty around the relationship increases when more assemblages are randomly removed, but the distribution of correlations remains positive and centred around 0.45-0.5.

(v) Finally, despite its bold claims, as to how rarity is changing within assemblages, the overall levels of extinction and immigration are small (1.9% and 3.5%, respectively). They are thus clearly not a major determinant of the compositional structure of these assemblages. Indeed, at these kinds of levels many might simply have dismissed the findings as rather uninteresting.

While we agree with the reviewer that the differences in overall levels of extinction and immigration of rare species by themselves is subtle, we disagree that they are uninteresting. We do so on three fronts.

1. The proportional difference between local extinctions and immigrations is small (1.6%), but this translated as a lot of species (>250)
2. The timescale over which these effects are happening may be important. Although these changes seem small, over time incremental changes can add up to substantial effects.
3. Our results by themselves may be less interesting, but they are another line of evidence building a clearer picture of how rarity is changing within the BioTIME assemblages.

We added the following text to Lines 210-214 to benchmark our results against the results of Dornelas et al 2015[15] so readers can compare more easily to what was found across all species abundance classes of assemblages and compare to actual numbers of species “Although the difference between the number of immigration and extinction events among rare species was not large (3.5% immigrations, 1.9% extinctions), it is still a larger proportional difference than the 8% immigrations vs 7 % extinctions detected by Dornelas et al. [15] when they assessed immigration and local extinctions in entire assemblages. Although the proportional differences are small, they relate to large numbers of species (>250 species).”

Reviewer: 2

Comments to the Author(s)

Several recent global studies found no consistent decreasing trends in local diversity, but relatively strong shifts in composition. Studies that look at how communities are changing are therefore key to understand such temporal patterns in species assemblages. This study focuses on how rare species contribute to the patterns of change, testing two main predictions (conceptualised in Fig. 1). The data were time series derived from the BioTIME database. I have one major comment and several smaller questions and suggestions, which generally focus on clarifying particular terms or quantitative approaches.

My main comment is that when going from pattern to process, one should carefully consider all possibilities that could create a certain pattern. On line 51 in the introduction, the authors say that “If declines in population sizes explain an increase in rarity, there should be no corresponding increase in species richness (Figure 1B)”. One possibility that is missing at this point, I believe, is a no net change in richness that is driven by population declines leading to local extinction and counterbalanced by immigration, that is, species turnover. The next paragraph seems to suggest this possibility, but I was a bit confused here because it is unclear whether Fig. 1B only represents declines in abundance without extinction (cf. figure caption) or also species turnover (as suggested on lines 56-58). So, it would help readers a lot if the text clarifies and justifies why the focus is on the scenario of species decreasing in abundance without going extinct only (see also abstract). The text should also be consistent in this context.

We thank the reviewer for rightly pointing out the inconsistency here in our schematic and process explanation. In response, we have substantially edited the text in various locations to clarify that some level of baseline turnover is expected, and that we are focusing on an increase in “net immigration” rather than “immigration”. We also changed the text in the abstract (Lines 10-12) to reflect this: “This increase could arise in two ways: species already present in the assemblage decreasing in abundance without increasing net extinction, or additional species entering the assemblage in low numbers increasing net immigration.”

We have added an additional scenario into our conceptual model where local extinctions increase (**Figure 1B/Figure R1**), and explained how an increase in immigration could mask such a situation.

We edited lines 41-48 to introduce the role of turnover in maintaining species richness and how increased net immigration may increase rarity “Immigration events maintain local species richness when immigration and local extinction events are balanced [10]. However, rates of turnover within assemblages are increasing [11]. This increase in turnover suggest a corresponding increase in immigration proportionally to local extinctions. Potential causes of this net increased immigration include climate change-driven range shifts [12] and human introductions [13]. Although most immigrant species fail to establish large local populations [14], their presence in low numbers may increase the number of rare species within assemblages if the number of immigration events exceeds the number of local extinctions.”

As mentioned above, to include local extinctions in our framework, we added to Line 36 the text “If the process of declining populations continues beyond species being retained due to extinction dept then this process will also eventually lead to declining local species richness as rarity increases” and to Line 35-39 the text “These local population declines may therefore be linked to increasing numbers of rare species within assemblages as more species populations fall below a threshold of rarity, assuming local extinctions do not increase. While rare species may be retained in the short term because of extinction dept, population declines could eventually lead to declining local species richness as rarity increases.”.

We also added the following text to the legend of **Figure 1**: “If there are local species extinctions due to population extinctions, there could either be decreases in species richness or local extinctions could be balanced against limited numbers of immigrations.”

Line 44: “two mechanisms” - I would rather use the term 'process' for the population declines and immigrations. The mechanisms driving these two processes can be very different and dependent on the species or systems where you look at.

Line 49 and 59 “mechanisms” changed to “processes”

Line 69: “summing the abundance of each species” - Doesn't it depend on the type of sampling whether or not taking a sum makes sense to remove the effect of seasonality? I'm working with plants and would never sum abundances when multiple records are available, because the same individuals are just counted multiple times then (especially for long-lived species that are just standing there). I can imagine that for many fauna communities, where individuals are actually captured and removed from the community, summing across sample occasions makes sense. We agree with the reviewer that summing abundances within years does not make sense for long lived sedentary individuals. While we have such plant communities in our analysis, these assemblage timeseries did not revisit the same sampling locations within years so we have not re-counted the same individuals. We have added the following text to Lines 103-105 to clarify this: “For faunal non-sessile studies with multiple samples per year, we also removed the effects of seasonality by summing the abundance of each species within each year. There were no floral or sessile faunal assemblages that had been resampled within years.”

Another point that should be clarified here as well, I think, is whether there are time series where the number of samples per year (if there are multiple) changes across time (e.g., more frequent within-year sampling in the most recent years). If so, this could also influence abundances when taking a sum across samples.

We agree that such variable sampling effort would likely influence results, but the sample based rarefaction process should account for changes in the number of samples per year. This point should be clearer now we have included more information on the rarefaction process in the main text and supplementary information (see response to Reviewer 1).

Line 86: “mean centred year” - Year was centred within or across time series? I assume within, so that intercepts represent the number of singletons and doubletons for the central year in a series. We have added the comment “where each year was centred around the mean year for the appropriate assemblage timeseries” after first mentioning mean centred years (Line 125) to clarify this

Line 88: the slope for global rarity change is not “calculated”, it is just the slope parameter of the year-effect in the statistical model? For the random slopes, you could say there are calculated, because in a model fitted with, for instance, lme they are not fitted as parameters.

We changed “calculated” to “estimated” in Line 127 to reflect this correction

Line 95: “strength and direction of the relationship between rates of change of rarity and rates of change of species richness for each study was then assessed” - So this is a correlation between the assemblage-level (i.e. per time series) slopes derived from the random effects of the previous mixed models? Please clarify. Also note that this analysis then ignores the uncertainty in these slopes, that is, time series in which the relationship with year comes with high uncertainty (std. error on the slope) are given equal weight as those in which the relationship is very tight. I know that models fitted with lme or similar functions cannot provide these std. errors and that it would require, for instance, MCMC methods to do so. But maybe the authors can simply acknowledge that the analysis does not account for uncertainty in the slopes, just to be clear about it. Essentially, this analysis is in fact a meta-analysis in which trends derived from independent assemblage time series data sets are compared, so readers may expect the dataset-level uncertainty to be accounted for.

The sentence in Line 39-143 has been updated as “The strength and direction of the relationship between rates of change of rarity and rates of change of species richness for each time-series was then assessed by extracting individual assemblage rates of change of change from the random effects of each model and assessing correlation using Pearson’s correlation coefficient.” We hope this clarifies how our analysis took place. The reviewer makes a very good point about including uncertainty around slope estimates, and is also correct that our models built in frequentist base packages did not include such data. We have reanalysed the data using models of the same structure but using a Bayesian approach in *brms* to account for variation in the assemblage level trends, and have updated the methods and result section accordingly. Our results were not modified by this methodological shift (**Figures S5-S7**). Because we saw only very small differences between the model outputs with and without second level variation, we chose to use the frequentist undertaking supplementary analysis of the effect of the rarefaction process and changes in numbers of singletons and Fisher’s alpha. Our decision was based on technical limitations, but we are confident that our results are consistent between modelling techniques.

Line 96: “study” – better to be consistent and refer to an assemblage or time series?

Line 141 “study” changed to “time-series”

Line 160: “biotic homogenization is a possible outcome of the increased immigration” - Only if it is a relatively small set of species that is immigrating across many sites. If the immigrants are a diverse set of species, each of which only immigrate in few sites, there will be no homogenization. Do the authors have evidence from their analyses for patterns of homogenization?

See below response

Line 162: “may differ in functional traits” – As with the homogenization, this is largely unknown from the results of this study, right? Together with the previous comment, I guess that future work should mainly focus on the identity and functional attributes of the species that immigrate and replace those that go locally extinct. [Note after reading the last paragraph: this aspect is actually covered in the conclusions; maybe integrate the lines 160-164 in that concluding paragraph?]

On reflection, we agreed with the reviewer, and have moved the text from lines 160-164 relating to biotic homogenisation to the conclusion paragraph Line 246-250. We believe this should avoid confusion by making it clear that functioning and immigrant identity is a potential avenue for further study rather than something covered in our analysis.

Appendix B

Reviewer comments to Author:

Reviewer: 2

Comments to the Author(s)

I have now carefully read the responses to my previous comments (referee 2) and am generally pleased with the way the authors handled them in preparing a revised manuscript. The authors' efforts to revise the conceptual diagram and to explain the possible scenario's more extensively throughout the text, will make the story much clearer for readers. There were some smaller points that I noticed when reading the manuscript again (I refer to line numbers of the manuscript version with track changes):

- Line 70-72 and Fig. 1A: "...if declines in population sizes without increases in local extinctions explain an increase in rarity, there should be no corresponding increase in species richness" Why linking local extinctions to increases in richness? It would be more logical if you say that there should be no changes in species richness (or no decreases in species richness, if you want to stress a direction). Also consider changing this in the text within Fig. 1A.

Line 70-71: Changed to "that increased immigration of species could cause no relationship between species richness and rarity like that predicted in Figure 1.1b if increasing immigration of rare species balances increasing species losses through local extinction." and in the text within Figure 1.

- Line 85: "...a balance of extinctions and colonisations..." In the rest of the text, the term immigration is used. Better to be consistent (and check throughout the entire manuscript)?

Changed colonisation to immigration on Line 79 and Line 168.

- Line 222: typo? "We detect increases a trend in increasing immigration..."

Changed to "We detected a trend of increasing immigration" in Line 213.

- Line 244: two typo's in one sentence? "...facet of biodiversity change in that abundant species may be being replaced by rarer ones."

Changed to "One previously unrecognised facet of biodiversity change is that abundant species may be being replaced by rarer ones."

In addition to these smaller points, I also noticed that referee 1 commented about the representativeness of the database for biodiversity change across the globe. Yes, the data are obviously not representing sites where wholesale habitat conversion took place, but to me this fact doesn't take away any relevance of the study for the posed questions. Temporal data are just not available for many of these heavily impacted sites and, maybe more importantly, it is just not relevant to look at changes in population sizes and how they scale to diversity changes in case a system is completely converted (virtually all species are lost and replaced by a complete set of other taxa). Besides describing more clearly what type of sites your study is focusing on (as you did in response to the comment of referee 1), I think it is also important to argue why the data are relevant for studying biodiversity change. Besides the arguments I mentioned, there are probably more ways to justify the use of this database for this particular study. Being clear about it should avoid strong criticism afterwards.

We have added to our methods section and substantially updated our discussion on the value of BioTIME (lines 87-96) to further emphasize the worth of BioTIME for such studies which ensuring caveats are clear. Our main points revolve around the fact that 1) there is no perfect dataset to tackle these sorts of large scale questions, 2) many assemblages in BioTIME are heavily affected by anthropogenic influences despite not being explicitly designed as before-after disturbance studies.

The methods section includes "The BioTIME database consists of assemblage time series in which sites have been monitored using a consistent methodology. Like all ecosystems on the planet, these assemblages are impacted by combinations of global stressors [17]. Specifically, BioTIME timeseries are affected by the ubiquitous climate change [18], marine data include locations affected by

overfishing [19], and most of the range in forest loss found across the planet is covered by BioTIME [20]. The widespread compositional change detected within BioTime assemblages is indicative of change captured by these data [21]. We note that our analysis applies only to the locations and time spans covered by these data, but these data are useful for gaining the nuanced overview of biodiversity change needed to avoid drawing overly simplistic results from few local studies [22].

The relevant discussion (Lines 213 -226) now reads: “We detected a trend of increasing immigration across over 100 assemblage studies from various taxa and realms. How general are these findings? As noted above, locations sampled in BioTIME, and consequently our analysis, include land- and seascapes affected by human drivers including overfishing, land-use intensification, and climate change [18–21]. Our results will, therefore, be relevant to a substantial fraction of global assemblages and provide an instructive indication of how such assemblages are changing. However, we recognise that the assemblages in our analysis are not a random subset of all ecological systems so caution is needed in interpreting the findings. The BioTIME database compiles data from systematically sampled assemblage time-series, and thus can only include assemblages that have already been monitored. This also means BioTIME does not include before-after-control-impact studies, and it is likely that habitats undergoing radical transformations due to severe impacts deviate from the mean trends detected in our models. A truly representative biodiversity change dataset is a key challenge for the future but unfortunately not yet available [22]. The analysis described in this paper does however benchmark changes in rarity for those assemblages for which quality time series data already exist.

Reviewer: 3

Comments to the Author(s)

I have received this manuscript for review after it having gone through one round of review. I have read the comments from the previous reviewers and believe that the issues they raised were appropriately dealt with, although I'll leave it to them and to the editor to make this decision. This is an interesting study which uses a very large dataset composed of 101 time-series to understand if patterns of rarity are changing over time, and if so why would that be the case. I enjoyed reading this, although I found a very large number of typos and awkward sentences that need to be thoroughly revised.

The explanation of species selection is confusing. In lines 107-118, it says that rare species were singletons and doubletons found within each year. Fine. Then in lines 147-149, the criterion for selecting rare species becomes “having a minimum population abundance of 1 or 2 within the assemblage”, so does this mean all species >1 individual, including the most abundant ones? So then you're excluding the rare species which are the focus of this ms, right? Now, assuming that selection criterion for rare species here is the same as lines 107-118, in which rare species are the singletons and doubletons. If you select only the rare species for this analysis, and ignore information of abundant species, does this mean that some colonisations and some extinctions may have been abundant species that became rare or rare species that became abundant (i.e., they were in and out of your selected dataset but not of the full dataset)? As you can see I'm very confused.

We thank Reviewer 3 for bringing to our attention the lack of clarity with regards to this section of the analysis and have added further clarification to lines 155-159: “Our selection criterion was population data of any species that had an abundance of one or two for at least one of the sampling years. This resulted in selecting 14,635 populations. We retained the within-species population dynamics of each selected species population, meaning population abundances in some years could rise above or fall below one or two individuals.

Of the issues raised by the previous reviewer, I would still like to highlight one. There were 518 cases of immigrations and 284 cases of extinction, so overall 234 new species over 101 assemblages. Of the populations tested, there were $(3.5-1.9)=1.6\%$ new species. I'm struggling to connect how such a small increase in number of species per timeseries/assemblage can lead to such strong increases in species richness. Assuming that these assemblages would have dozens of species, it's hard to see how one extra species would lead to a strong increase in richness over time. The authors included a new part of discussion to explain this, but I confess that it didn't fully convince me, and this may be in part because I didn't understand the immigration vs extinction analyses (comment above). Or maybe what needs to be done is to reassess whether the increases in richness are in fact strong, or just barely significant (as lower 95%HPDI is 0.0007).

We agree that our description of the immigration/extinction part of the analysis was confusing, leading to the conclusion that we include as immigration/extinction events abundant species becoming rarer or rarer species becoming abundant. This was not the case, and to avoid such confusion in the future we have further clarified our description of the analysis (see comments above). We agree with Reviewer 3 that if we had included such transitions between rare/abundant states in our analysis of immigrations and extinctions then our results would be far less convincing.

We will now address Reviewer 3's other concerns and suggestions regarding this analysis. Although the lower 95% HPDI is only 0.0007, we argue that this does not mean the increase in species richness is barely significant. Our analysis used \log_{10} transformed species richness values, so in our discussion, we consider the slope values in real terms (an increase of roughly 1 species per year on average). The length of the timeseries of our assemblages analysis vary but are generally between 10 -20 years long. This means our positive slope does not suggest large increases in species richness during the time period of the assemblages, especially when some assemblages are experiencing decreasing species richness. Although this increase in species richness is not large, such a trend has the potential to lead to large future changes if it continues. We acknowledge though that the detected increase in rare species immigrating rather than going locally extinct does not fully explain the increase in species richness. Increasing numbers of rare species immigrating into assemblages are unlikely to be the only factor influencing changes in species richness.

We have modified the discussion on Lines 231 -237 by adding the following text: "In addition although the trend we detected in species richness across the 101 assemblages has a very high probability of being positive (95% HPDI between 0.0007 and 0.0043), our analysis used a \log_{10} transformation. This means the estimated mean slope value translates to an increase on average of 1 species per year ($100.0007=1.002$, $100.0043 = 1.010$) which is in the same order of magnitude as the difference between the number of species immigration and going extinct from our analysis of population trends. Increasing rare species immigration is also unlikely to be the only influence on species richness."

Line 166 – here it says that the increase in rarity was more pronounced than using Fisher's alpha, but that's not what Fig S9 shows. SE overlaps 0.

This was an error on our part. The text in Line 176 has now been corrected to reflect the fact that the Fisher's alpha model does not show a significant trend. "There was also a positive trend detected using Fisher's Alpha (Figure S9), but this trend was not as strong."

The text in Lines 199-201 has also been modified to reflect the correction: "This result did not change when rarity was defined as the number of singletons (Figure S8), but it was less clear when evaluated using Fisher's alpha (Figure S9)."

Fig 3 – inclusion of panel C to respond to Reviewer 1 seems out of context here. I would suggest including this in the hypotheses, or as part of the trends you expect to find, because otherwise this will make little sense for further readers.

On the suggestion of Reviewer 3, and to increase clarity, we have made extensive changes to the text introducing our processes and hypothesis. This helps justify the inclusion of **Figure 3C** and makes our predictions regarding the different processes easier to follow. To this end, we have revised the text where we introduce the different scenarios (lines 49-82). To make it clearer we have numbered the two different processes as process 1) *decreasing immigration* and process 2) *increasing immigration*. We hope that this labeling will make it easier for the reader to keep track of the different processes, and outcomes of the processes.

We also added text to specify relationships between changes in species richness and assemblage size to give Figure 3C more context.

Lines 58 to 69: “If process 1) decreasing population sizes dominates, we foresee two potential outcomes. In both outcomes, rarity is caused by a decline in population sizes of species already present. In the first of the two outcomes, however, there is no corresponding increase in net local extinctions and so we expect no change in species richness, a decline in assemblage size, and no relationship between changes in species richness and assemblage size (Figure 1.1a). The alternative outcome for process 1) decreasing populations will arise if there is an increase in net extinction with increasing rarity. In this case, we expect co-varying negative trends in species richness and assemblage size (Figure 1.1b). In both outcomes, assemblage size should decline with increasing rarity due to fewer individuals being present in the assemblage.”

Lines 67-69: “In process 2) *increasing immigration*, where increasing rarity is mainly driven by an increase in net immigration, we expect trends in rarity, species richness, and assemblage size to all be positive and co-vary [13](Figure 1.2). It is worth noting that increased immigration of species could cause no directional relationship between species richness and rarity like that predicted in **Figure 1.1b** if increasing immigration of rare species balances increasing species losses through local extinction. The relationship between rarity and assemblage size would differ, though, as there would be no directional relationship between trends in these facets of biodiversity, nor a directional relationship between trends in assemblage size and species richness.”

We have also condensed our introduction. This means that the two processes are introduced in the context of the relevant panels of our conceptual diagram (Figure 1). In addition, the labeling of Figure 1 is now in line with Process 1 and 2 rather than Scenario A, B, and C.

Appendix C

Comments to the reviewers

Comments to the Author(s)

The authors have carefully addressed my previous questions. Especially the updated sections in the methods and discussion on the representativeness and value of BioTIME for the research question of this study are a valuable addition to the manuscript. Note that I have only evaluated the changes that were made in response to my previous review and have not looked in detail at the many edits in the text that were made in response to reviewer 3.

We thank the reviewer for their previous comments and are pleased that they are satisfied with them.

Reviewer: 3

Comments to the Author(s)

I appreciate the authors' responses to my comments. I am much happier with the text now. I do have just one question for the authors and would like them to consider whether including a sentence or two in the discussion would be justifiable. Do you think that tracking species identity for all these communities through time would have been feasible and would have resulted in a better estimation of whether the changes in rarity are caused by a mix of immigration and extinction? Because, as it's mentioned in the paper, the two processes are not mutually exclusive and my feeling from the methods is that they can't completely set apart the relative importance of extinction or population reductions and immigrations. Maybe I got this wrong and the authors are indeed doing that, in which case again adding a sentence about clarifying this would improve clarity.

We are glad that Reviewer 3 is generally happy with our revisions and have addressed their final comments. See below for details.

We chose to focus on assemblage structure in this analysis because this is something that is comparable between different community types and so our results would be most generalisable. We agree, though, that information on specific species populations is also useful for understanding rarity change. The second part of our analysis, where we estimated numbers of rare species going locally extinct or immigrating, did track a subset of individual species populations and provide some extra detail on the drivers of increasing rarity. The type of analysis that Reviewer 3 is talking about, where one tracks how the population of every species within the assemblages would definitely be useful to provide further clarity on the prevalence of immigration and extinction events, but is outwith the remit of our specific analysis. Such an analysis is feasible with sufficient time and thought, though, and would make a logical follow up from our more structurally focused work.

To ensure this point is covered in our manuscript, we have added the following text to our discussion:

Line 206 -210: "Our results, though, do not suggest that process 1) decreasing populations are not also contributing to increasing rarity, but that process 2) increasing immigration is the main driver of detected changes. Further analysis focusing on tracking individual species populations within assemblages is required to elucidate the proportion species experiencing population declines vs immigration events."

Line 246 – change "may be being" to "are maybe being"

Corrected.

References

Dornelas, M. *et al.* (2019) 'A balance of winners and losers in the Anthropocene', *Ecology Letters*, 22, pp. 847–854. doi: 10.1111/ele.13242.